# A Comparative Study of Gamma-Ray Irradiation-Induced Oxidation: Polyethylene, Poly (Vinylidene Fluoride), and Polytetrafluoroethylene

**DOI:** 10.3390/polym14214570

**Published:** 2022-10-28

**Authors:** Ha-Eun Shim, Byoung-Min Lee, Dae-Hee Lim, You-Ree Nam, Pyung-Seok Choi, Hui-Jeong Gwon

**Affiliations:** 1Advanced Radiation Technology Institute, Korea Atomic Energy Research Institute, 29 Geumgu-gil, Jeongeup 56212, Korea; 2Department of Food and Nutrition, Chungnam National University, 99 Daehak-ro, Yuseong-gu, Daejeon 34134, Korea

**Keywords:** irradiation, polyethylene, polyvinylidene fluoride, polytetrafluoroethylene, degradation, oxidation

## Abstract

Radiation techniques are used to modify the physical, chemical and biological properties of polymers. This induces crosslinking and degradation reactions of polymers by utilizing radicals generated through ionizing radiation. However, oxidation products (such as carbonyl) can be formed because oxidation occurs by chain scission in the presence of oxygen. Herein, we demonstrate the gamma-ray irradiation-induced oxidation with and without fluorine using polyethylene, polyvinylidene fluoride and polytetrafluoroethylene under the same conditions. In this study, changes in element-content and chemical-bond structures were analyzed before and after gamma-ray irradiation under air atmosphere. As a result, polytetrafluo-roethylene showed less oxidation and excellent thermal properties after the absorbed dose of 500 kGy. This can be attributed to the generation of stable perfluoroalkylperoxy radicals after gamma ray irradiation in the PTFE structure containing only CF_2_ groups, thereby hindering the oxidation reaction.

## 1. Introduction

Radiation technology utilizes ionizing radiations to modify and/or improve the physical, chemical, and biological properties of polymers [1,2,3,4]. Ionizing radiations (including electron beams, gamma-rays, and proton beams) are used for the surface modification, crosslinking, curing, and degradation reactions of polymers [5,6,7,8,9]. Polymer reactions utilize radicals that are generated on exposure to ionizing radiation; they can be classified as crosslinking reactions (in which the molecular weight increases due to bonds between the polymer chains) and degradation reactions (in which the molecular weight decreases due to polymer-chain cleavage) [7].

The ionizing-radiation method exhibits several advantages compared to other traditional methods for inducing polymer-material reactions; it does not require chemical additives and induces reactions rapidly at room temperature [10,11,12]. Irradiation in vacuum or inert atmosphere promotes the crosslinking reaction, while chain scission and oxidative degradation predominantly occur in air–atmosphere irradiations [13,14]. In addition to the crosslinking and degradation reactions, carbonyls and other oxidation products are formed in the presence of oxygen [14,15].

Polyethylene (PE) is used largely in the radiation-technology industry; therefore, the effects of ionizing radiation on it have been extensively studied [16,17,18]. Fluoropolymers exhibit excellent chemical/thermal properties and weathering due to the high bonding energy of carbon and fluorine, and the high values of fluorine electronegativity and electron density [19,20,21]. On irradiating the per-fluorinated polymers polytetrafluoroethylene (PTFE), polychlorotrifluoroethylene (PCTFE), poly(tetrafluoroethylene-co-hexafluoropropylene) (FEP), and poly(tetrafluoroethylene-co-perfluoro(propyl vinyl ether)) (PFA) with ionizing radiation under air atmosphere, degradation occurs via chain scission [22,23,24]. On the other hand, the crosslinking and degradation reactions simultaneously occur upon irradiating partially fluorinated polymers containing hydrogen, carbon, and fluorine in their main chains (such as, poly(vinylidene fluoride) (PVDF), poly(vinyl fluoride) (PVF), poly(trifluoroethylene) (PTrFE), and ethylene-co-tetrafluoroethylene (ETFE)) with ionizing radiation under air atmosphere [25]. Partially fluorine-based polymers exhibit a higher radiation resistance compared to fluorine-based polymers that undergo degradation under ionizing radiation; this could be because crosslinking prevails over degradation in the former due to the radicals generated in the C-H structure of the main chain [26].

In this study, PE, PVDF, and PTFE were used to compare and analyze the effects of gamma rays on structurally different polymers with and without fluorine. PE, a simple polymer composed of hydrocarbons, can easily be compared with a partially fluorinated polymer (PVDF) and a perfluorinated polymer (PTFE).

Coote et al. performed gamma irradiation with absorbed doses between 30.0 and 44.1 kGy after packaging polyethylene sheets at a pressure of 18 mbar [27]. Batista et al. observed the oxidation reaction by exposing the PVDF membrane to an absorbed dose of up to 3000 kGy under air atmosphere [28]. Saidi-Amroun et al. compared the degradation of PTFE films induced by gamma rays with absorbed doses of 60 and 600 kGy at an irradiation dose rate of 22 Gy/min under air atmosphere [29]. As far as we know, it is difficult to compare the oxidation reaction of three polymers simultaneously under the same conditions due to the different equipment and different conditions used.

Here, gamma-ray irradiation under air atmosphere was used to investigate the oxidation reaction in the structurally different PE, PVDF, and PTFE. The elemental composition and chemical structure of PE, PVDF, and PTFE after gamma-ray irradiation and the consequent changes in their chemical structure were compared and analyzed. Furthermore, changes in their thermal and physical properties were investigated and their oxidation-reaction mechanisms were discussed.

## 2. Materials and Methods

### 2.1. Sample Preparation and Radiation

Polyethylene (PE, density: 0.94 g/mL), poly(vinylidene fluoride) (PVDF, density: 1.74 g/mL, Mw: ~534,000), and polytetrafluoroethylene (PTFE, density: 2.15 g/mL) were purchased from Sigma-Aldrich (St. Louis, MO, USA). The three powder-type polymers were used at the high-level gamma-ray irradiation facility (MDS Nordion, IR 221n wet storage type C-188) at the Advanced Radiation Research Institute of the Korea Atomic Energy Research Institute at an irradiation dose rate of 10 kGy/h under air atmosphere using a ^60^Co source. The absorbed doses were 25, 50, 100, 200, and 500 kGy.

### 2.2. Characterization

Fourier-transform infrared (FT-IR) spectrometry (Prestige-21, Shimadzu, Kyoto, Japan) and X-ray photoelectron spectroscopy (XPS, K-alpha+, Thermo Fisher Scientific, Waltham, MA, USA) were used for the elemental analysis and to investigate changes in the chemical bonding and structure. An X-ray diffractometer (XRD, D8 DISCOVER, Bruker AXS, Billerica, MA, USA) was used to analyze crystallinity changes in the range of 2θ = 5–60°. The *d*-spacing was calculated using Bragg’s law [30], where n is the reflection order, *λ* is the wavelength of X-ray radiation (0.154 nm), *d* is the interplanar distance, and the θ is the Bragg’s angle of incidence.
*nλ* = 2*d*sinθ(1)

The degree of crystallinity (*X_c_*) was calculated using the following equation [31]:(2)$$X_{c}\left(\%\right)=\frac{A_{c}}{A_{c}+A_{a}}\times 100$$
where *A_c_* is the area of total crystalline region and Aa is the area of total amorphous region. A thermogravimetric analyzer (TGA, SDT Q600, TA Instruments, New Castle, DE, USA) was used to investigate the thermal-property changes in the temperature range of 25–800 °C by increasing the temperature at a rate of 10 °C/min under nitrogen atmosphere. Differential scanning calorimetry (DSC, SDT Q600, TA) was used to monitor the change in the melting temperature (T_m_) by increasing the temperature from room temperature to 350 °C at a rate of 5 °C per minute under nitrogen atmosphere.

## 3. Results and Discussion

### 3.1. ATR-FTIR

Figure 1 shows the ATR-FTIR results indicating the changes in the chemical-bond structures of PE, PVDF, and PTFE according to the absorbed doses of gamma rays. PE exhibited characteristic peaks for CH_2_ binding at 2914 cm^−1^ (asymmetric stretching), 2848 cm^−1^ (symmetric stretching), 1463 cm^−1^ (bending deformation), and 719 cm^−1^ (rocking deformation) (Figure 1a), while PVDF exhibited CH_2_ binding peaks at 3024, 2980, and 1402 cm^−1^. Also, PVDF showed characteristic peaks for CF groups (1180 and 840 cm^−1^) (Figure 1c) [32,33]. For PTFE (Figure 1e), characteristic peaks in the CF_2_ structures were observed at 1201 cm^−1^ (asymmetric stretching) and 1145 cm^−1^ (symmetric stretching), respectively [34,35]. After irradiation with gamma rays, a characteristic carbonyl (C=O) peak was observed at 1720 and 1726 cm^−1^ in PE and PVDF, respectively (Figure 1b,d); this could be due the reaction between gamma-ray irradiation-generated radicals and aerial oxygen [16,17]. The oxidation reaction was initiated by the absorption of a lower dose of gamma rays in PE than in PVDF; this could be due to the presence of a greater number of CH_2_ structures in PE compared to PVDF. However, PTFE, unlike PE and PVDF, did not exhibit a C=O bond structure; this could be due to the absence of oxidation-reaction-facilitating C-H structures. Additionally, after irradiation with an absorbed dose of 500 kGy, it exhibited a characteristic peak at 1780 cm^−1^, indicating the COO-bond structure. Thus, the C-F bond in the PTFE structure exhibited low reactivity with radicals generated by the absorption of high absorbed doses, hindering facile oxidation [36,37]. The FTIR results confirmed a facile oxidation of the C-H bond structure in PE and PVDF due to hydroperoxide radicals generated by the reaction of gamma-ray irradiation-generated radicals with aerial oxygen [38,39]. PTFE exhibited the lowest oxidation on gamma-ray irradiation among all the analyzed polymers; this could be attributed to the generation of stable perfluoroalkylperoxy radicals in the PTFE structure on gamma-ray irradiation, which hindered the oxidation reaction [40].

### 3.2. XPS Analysis

Figure 2 shows the XPS analyses of changes in the C, O, and F content of PE, PVDF, and PTFE according to the absorbed doses of gamma rays. In the XPS survey spectra (Figure 2a–c), the peaks in C, O, and F were observed at 285, 532, and 680 eV, respectively; Figure 2d–f show the element-content change according to the absorbed doses of gamma rays, as indicated by the XPS analysis. In Figure 2d, a small amount of O before gamma-ray irradiation was measured by XPS analysis under air atmosphere. The O content of PE and PVDF increased with an increase in the absorbed dose, indicating increased oxidation. However, the small change in the O content of PTFE compared to that of PE and PVDF indicated limited oxidation in PTFE. PE exhibited an O content of 1.16% on the absorbed dose of 25 kGy. In Figure 2d, after irradiation with 500 kGy of absorbed dose, the O contents of PE, PVDF, and PTFE were 2.39, 1.7, and 0.47%, respectively; the values were unexpectedly low due to the presence of CF-bonds in the structures of PVDF and PTFE. According to the C content analysis results shown in Figure 2e, PE, PVDF, and PTFE without gamma-ray irradiation exhibited C contents of 99.61, 52.85, and 34.72%, respectively. The C contents decreased as the absorbed dose increased; the C contents decreased to 97.61, 52.52, and 34.81% for PE, PVDF, and PTFE, respectively, after irradiation with a dose of 500 kGy of absorbed gamma-rays. Additionally, PVDF and PTFE exhibited F contents of 46.68 and 64.84%, respectively, which decreased on increasing the absorbed dose (Figure 2f). After irradiation with an absorbed dose of 500 kGy, the F content of PVDF and PTFE decreased to 45.78 and 64.72%, respectively. Therefore, the O and C content ratio ([O]/[C] ratio) of PE increased from 0.0039 to 0.0244, while the F and C content ratio ([F]/[C] ratio) of PVDF and PTFE decreased from 0.883 to 0.871 and 1.867 to 1.859, respectively, after increasing the absorbed dose. This reduction in F content could be attributed to the defluorination of PVDF and PTFE by gamma-ray irradiation [41,42].

Figure 3 shows the deconvolution of the XPS C1s narrow spectra that was used to analyze changes in the chemical-bond structures of PE, PVDF, and PTFE before and after irradiation with an absorbed dose of 500 kGy. PE without gamma-ray irradiation (labeled PE_0 kGy) exhibited the characteristic peak of the C-C bond (285.08 eV) [43], PVDF without gamma-ray irradiation (labeled PVDF_0 kGy) exhibited characteristic peaks corresponding to C-C bonds (285.1 eV) and CF bonds (289.6 eV) [44], while PTFE without gamma-ray irradiation (labeled PVDF_0 kGy) exhibited a characteristic peak indicating CF_2_ bonding (291.08 eV) [45]. PE and PVDF were labeled PE_500 kGy and PVDF_500 kGy, respectively, after gamma-ray irradiation of 500 kGy, and were oxidized by radicals generated in their structures to form C-O/C=O (286.28 eV) and C=O bonds (287.18 eV), respectively [46]. Element-content and chemical-bond structural-change analyses using XPS confirmed that the O content of PE and PVDF increased as the absorbed dose increased, generating C-O/C=O and C=O bonds, respectively. Furthermore, the CH and CF bonds were decomposed through dehydrogenation and defluorination, respectively, due to gamma-ray irradiation [47,48,49]. Additionally, PTFE containing only CF_2_ groups was not easily oxidized by gamma-ray irradiation. These results based on the XPS analysis are consistent with the FTIR results shown in Figure 1.

### 3.3. XRD Analysis

XRD analysis was used to investigate changes in crystallinity according to the absorbed dose of gamma rays (Figure 4). PE exhibited characteristic peaks for orthorhombic planes (110) and (200) at 2θ = 20.6° and 23.7°, respectively [50]. PVDF exhibited characteristic peaks corresponding to the (020), (110), and (021) planes at 2θ = 18.4°, 19.9°, and 26.6°, respectively. The peaks at 2θ = 18.4° and 26.6° was attributed to the α-phase of PVDF and the peak at 2θ = 19.9° was related to the β-phase of PVDF [51]. In the case of PTFE, the characteristic peak in the (100) plane at 2θ = 18.0° was detected [52]. These characteristic peaks for PE, PVDF, and PTFE at 2θ = 20.6°, 19.9°, and 18°, respectively, were shifted slightly to lower 2θ angle with an increase in absorbed doses. Before gamma-ray irradiation (0 kGy), the *d*-spacing values of PE, PVDF, and PTFE were 0.427, 0.476, and 0.439 nm, respectively. These *d*-spacing values increased to 0.431 nm (PE), 0.499 nm (PVDF), and 0.521 nm (PTFE) after gamma-ray irradiation of 500 kGy, respectively. Figure 4d shows the changes in crystallinity according to the absorbed dose of gamma rays. The decrease in the crystallinity as the absorbed dose increased is due to an increase in the relatively amorphous region after the gamma-ray irradiation. Based on XRD analyses, these results expected that oxidation reactions in polymer structures due to irradiation primarily occur in the amorphous region; the crystallinity decreases because the oxygen-containing functional group that is generated interferes with the regular arrangement of the polymer chain.

### 3.4. Thermal Analysis

Changes in the thermal properties of PE, PVDF, and PTFE due to gamma-ray irradiation were analyzed by TGA and DSC (Figure 5). As shown in the DTG curve in Figure 5a–c, before gamma-ray irradiation, PE (0 kGy) and PVDF (0 kGy) underwent thermal decomposition in the temperature range of 390–500 °C and 430–500 °C, respectively, while PTFE (0 kGy) underwent thermal decomposition in the temperature range of 510–600 °C. Before gamma-ray irradiation, the decomposition temperatures (T_d_) were 474.8, 484.3, and 576.1 °C for PE, PVDF, and PFTE, respectively. These values decreased to 461.6, 457, and 570 °C, respectively, upon irradiation with an absorbed dose of 500 kGy of gamma rays. As shown in the DSC curve in Figure 5d–f, the Tm of PE, measured by DSC analysis, were 109.5 and 112.5 °C; these values decreased to 108.9 and 111.4 °C, respectively, after irradiation with an absorbed dose of 500 kGy. For PVDF, the T_m_ of 156.93 and 161.53 °C decreased to 149.95 and 156.04 °C, respectively, after irradiation with 500 kGy of absorbed dose. For PTFE, the Tm before irradiation was 323.67 °C; this decreased to 320.61 °C after irradiation with an absorbed dose of 500 kGy. Thus, the thermal properties of PE, PVDF, and PTFE (such as T_d_ and T_m_) decreased after irradiation with gamma rays. Thermal properties related to the degradation could be attributed to the structural decomposition (of the main chain) due to oxidation [53,54,55]. Therefore, among the analyzed polymers, PTFE, with the highest fluorine content, underwent the least oxidation and decomposition by gamma-ray irradiation, and exhibited the highest radiation resistance.

## 4. Conclusions

In this study, we discuss the gamma-ray irradiation-induced oxidation of PE and fluoropolymers (PVDF and PTFE) under air atmosphere. After gamma-ray irradiation, the thermal properties of PE, PVDF, and PFTE were decreased, which is considered to be decomposition due to oxidation. However, among the three polymers, PTFE showed the lowest oxidation in the analyzed gamma ray irradiation as a result of FTIR, XPS, and XRD analysis. This can be attributed to the generation of stable perfluoroalkylperoxy radicals in the PTFE structure against gamma irradiation, which interferes with the oxidation reaction. Therefore, it was confirmed that PTFE including only CF_2_ groups had the highest fluorine content and the lowest oxidation and decomposition by gamma irradiation.

## Figures and Tables

**Figure 1 polymers-14-04570-f001:**
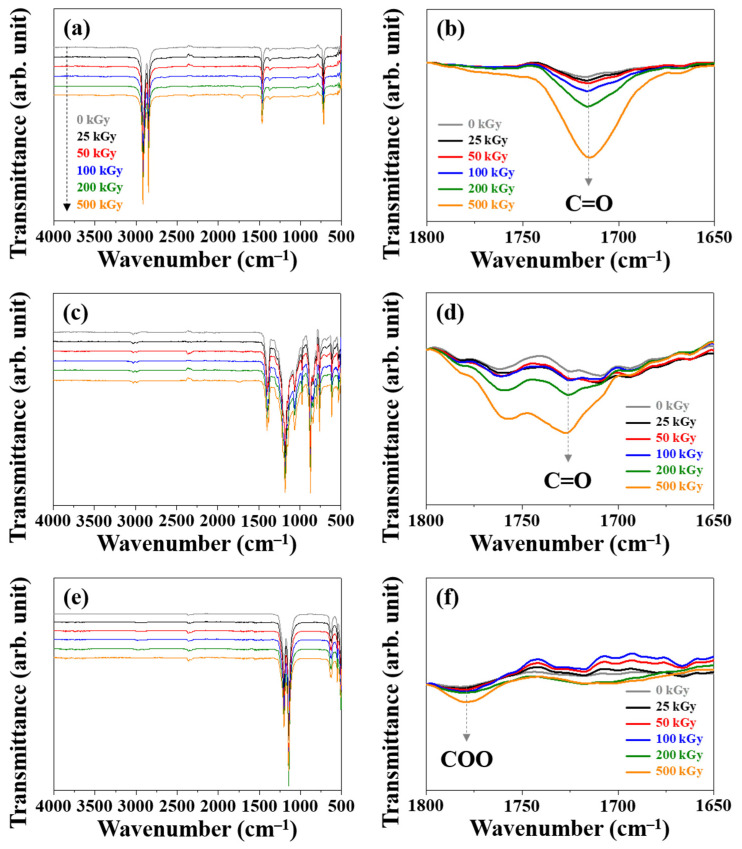
FTIR spectra of (**a**,**b**) PE, (**c**,**d**) PVDF, and (**e**,**f**) PTFE.

**Figure 2 polymers-14-04570-f002:**
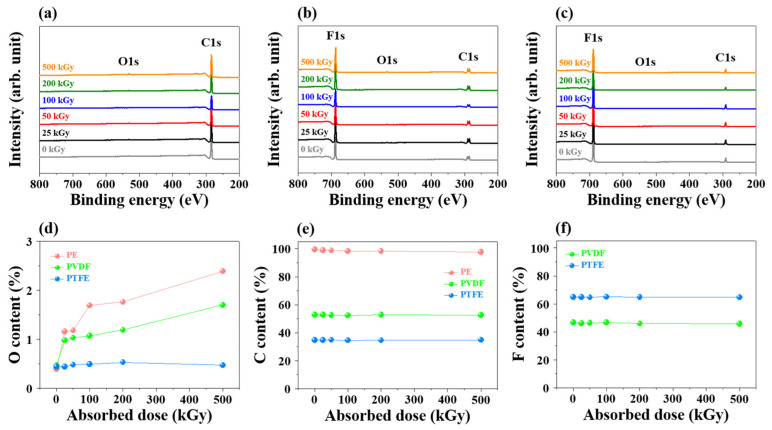
XPS survey spectra of (**a**) PE, (**b**) PVDF, and (**c**) PTFE; Changes in the atomic percentages of (**d**) oxygen, (**e**) carbon, and (**f**) fluorine.

**Figure 3 polymers-14-04570-f003:**
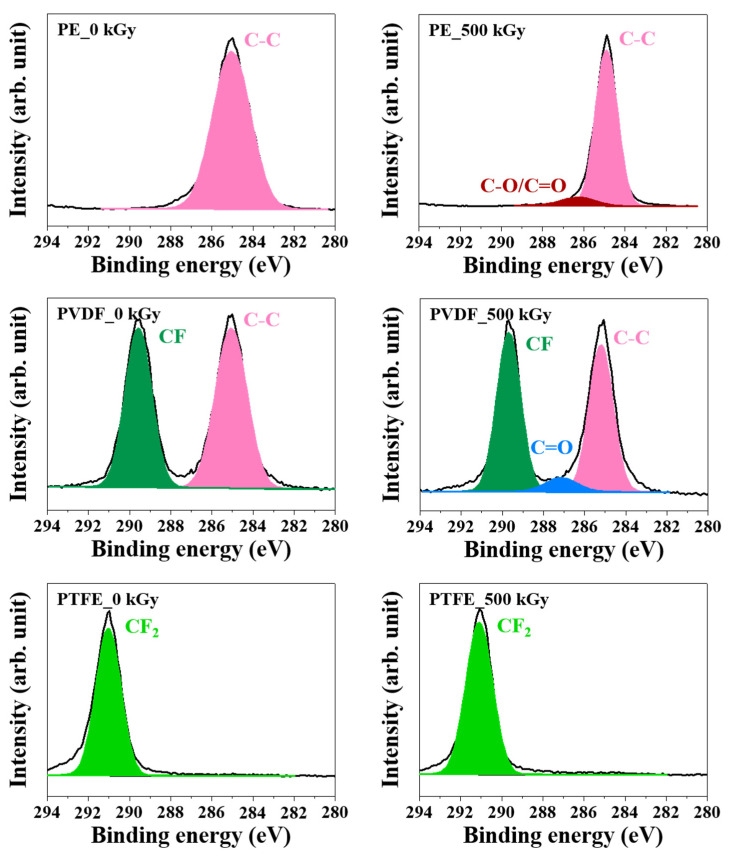
The deconvoluted XPS C1s narrow spectra of PE, PVDF, and PTFE, before and after gamma-ray irradiation.

**Figure 4 polymers-14-04570-f004:**
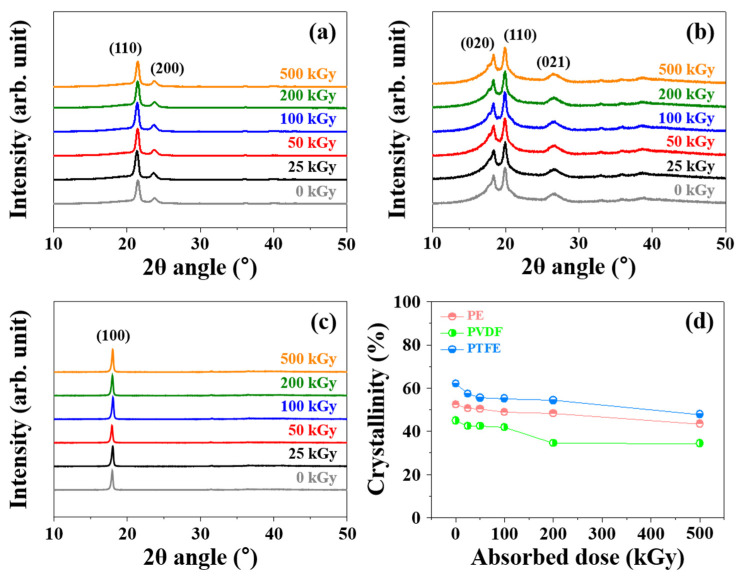
XRD patterns of (**a**) PE, (**b**) PVDF, and (**c**) PTFE; (**d**) Changes in the crystallinity of PE, PVDF, and PTFE.

**Figure 5 polymers-14-04570-f005:**
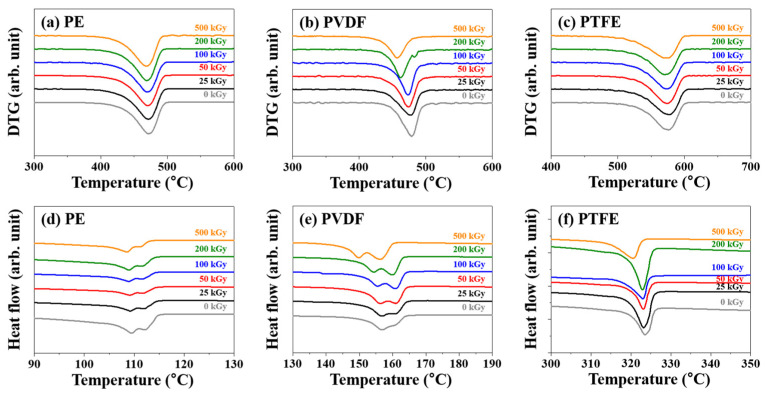
DTG curves of (**a**) PE, (**b**) PVDF, and (**c**) PTFE; DSC curves of (**d**) PE, (**e**) PVDF, and (**f**) PTFE.

## Data Availability

Not applicable.

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
