# Peer review of "A Comparative Study of Gamma-Ray Irradiation-Induced Oxidation: Polyethylene, Poly (Vinylidene Fluoride), and Polytetrafluoroethylene"

_polymers, 2022, doi:10.3390/polym14214570_

Round 1
Reviewer 1 Report
The authors analyzed the PE, PVDF and PTFE before and after gamma ray irradiation using the FTIR, XPS, XRD and TG/DSC. These works can be found in many references. In addition, not any new results/information and conclusion can be found in this manuscript.
lines 77-80 should be removed.
After gamma irradiation in air, the scission and crosslinking as well as oxidiaiton occurred. as results, the chemical, thermal and mechanical properties should be detected. Although the authors concluded that the ptfe was less oxication, it is well known tha the mechanical strenghth of the PTFE was largely decreased after irradiation in air.
Author Response
The authors analyzed the PE, PVDF and PTFE before and after gamma ray irradiation using the FTIR, XPS, XRD and TG/DSC. These works can be found in many references. In addition, not any new results/information and conclusion can be found in this manuscript.
Point 1. lines 77-80 should be removed.
Response 1: The correction was carried out in the revised report as suggested.
Point 2. After gamma irradiation in air, the scission and crosslinking as well as oxidiaiton occurred. as results, the chemical, thermal and mechanical properties should be detected. Although the authors concluded that the ptfe was less oxication, it is well known tha the mechanical strenghth of the PTFE was largely decreased after irradiation in air.
Response 2: It is known that the influence changes depending on the condition of the material and the irradiation conditions. For example, when PFTE is irradiated in a molten state, the fluidity of the chain increases and is crosslinked (reference).
- Watari, K.; Toru I.; Motoshige Y. Structure change of PTFE by low‐energy ion irradiation. Restraint of structure collapse by crosslinking structures. Electrical Engineering in Japan 2012, 178, 1–7.
- Tamada, M. Radiation processing of polymers and its applications. Radiation applications. Springer, Singapore, 2018; 63–80.
Our study investigated only the oxidation reaction by simple irradiation in an air atmosphere. Therefore, although the current research is a simple comparison, we will present a study that can improve radiation resistance for each material in the future work.

Reviewer 2 Report
This manuscript reports on the study of the oxidation induced by gamma ray irradiation on different polymers. Irradiation with gamma ray of polymers (and fluoropolymers) is not new so it should be clearly stated which is the novelty of this work because it is not clear. At the same time, experimental results should be compared to literature.
Besides this important point other aspects to be clarified/improved are:
- Information is missing in the materials and methods section. In particular, regarding the polymers, it is mentioned where they were purchased but no details are given about whether the polymers were treated/manipulated somehow or if used as received. In such a case where they were placed for irradiation, how, in what amount, was the irradiation homogeneous, etc.
- In the characterization section (2.2) I guess the first paragraph should be deleted.
- Is the molecular weight of the polymers modified after irradiation, since chain scission is occurring?
- In the discussion or in the conclusions: according to the results obtained what is the interest of doing this studies and of irradiating with gamma radiation these polymers? What would be the effect on applications or on other properties of interest of these materials? It had been reported that for instance for PTFE not only the chemistry is modified but also the mechanical properties are worse after this kind of treatment.
Author Response
This manuscript reports on the study of the oxidation induced by gamma ray irradiation on different polymers. Irradiation with gamma ray of polymers (and fluoropolymers) is not new so it should be clearly stated which is the novelty of this work because it is not clear. At the same time, experimental results should be compared to literature.
Besides this important point other aspects to be clarified/improved are:
Point 1. Information is missing in the materials and methods section. In particular, regarding the polymers, it is mentioned where they were purchased but no details are given about whether the polymers were treated/manipulated somehow or if used as received. In such a case where they were placed for irradiation, how, in what amount, was the irradiation homogeneous, etc.
Response 1: Reagents were used as they were without pre-treatment. The powder was not sealed by putting 5 g of each in a vial, and it was irradiated in an open state.
Point 2. In the characterization section (2.2) I guess the first paragraph should be deleted.
Response 2: The correction was carried out in the revised report as suggested.
Point 3. Is the molecular weight of the polymers modified after irradiation, since chain scission is occurring?
Response 3: It is correct that molecular weight decreases when chain scission occurs (ref. 1). On the other hand, it is known that the molecular weight increases according to the crosslinking reaction (ref. 2).
- Speight, J.G. Handbook of Industrial Hydrocarbon Processes, 1st ed.; Elsevier/Gulf Professional Publ: Amsterdam, The Netherlands, 2011; ISBN 978-0-7506-8632-7.
- Triwulandari, E.; Fatriasari, W.; Iswanto, A.H.; Septiyanti, M.; Umam, E.F.; Ghozali, M. Effect of Reaction Time on the Molecular Weight Distribution of Polyurethane Modified Epoxy and its Properties. Mater. Res. Technol. 2022, 19, 2204–2214
Point 4. In the discussion or in the conclusions: according to the results obtained what is the interest of doing this studies and of irradiating with gamma radiation these polymers? What would be the effect on applications or on other properties of interest of these materials? It had been reported that for instance for PTFE not only the chemistry is modified but also the mechanical properties are worse after this kind of treatment.
Response 4: Research results have been reported only for each material so far, but it is judged that there is no study result comparing the oxidation reaction by radiation under the same conditions. Therefore, this study comparatively studied the oxidation of three polymers to gamma rays in the air at the same time. When PTFE is irradiated in the melt state, the fluidity of the chain increases and crosslinks (ref. 3,4).
- Watari, K.; Toru I.; Motoshige Y. Structure change of PTFE by low‐energy ion irradiation. Restraint of structure collapse by crosslinking structures. Electrical Engineering in Japan 2012, 178, 1–7.
- Tamada, M. Radiation processing of polymers and its applications. Radiation applications. Springer, Singapore, 2018; 63–80.
Therefore, although it is a simple comparison for now, we will present a study that can improve the radiation tolerance for each material in the future at the next opportunity.

Reviewer 3 Report
Work submitted on gamma-ray irradiation-induced oxidation of three different polymers namely, polyethylene, poly(vinylidene fluoride), and polytetrafluoroethylene; is interesting and can be published in the journal. However, I have a few suggestions
Modify the abstract to attract the readers.
Why authors did not study the mechanical properties? As radiation significantly reduced the strength of polymeric materials.
Thermal properties are also missing. Authors must study the behavior of the polymers before and after irradiation.
Author Response
Work submitted on gamma-ray irradiation-induced oxidation of three different polymers namely, polyethylene, poly(vinylidene fluoride), and polytetrafluoroethylene; is interesting and can be published in the journal. However, I have a few suggestions.
Point 1. Modify the abstract to attract the readers.
Response 1: The mentioned correction has been carried out and incorporated into the text in the revised report.
Point 2. Why authors did not study the mechanical properties? As radiation significantly reduced the strength of polymeric materials.
Response 2: Previously, studies have been reported in which the mechanical properties of the three polymers decreased as the irradiation dose increased due to the decomposition of polymer chains when gamma-irradiated under atmospheric conditions.
- Ferreto, H.F.R.; Oliveira, A.C.F.; Gaia, R.; Parra, D.F.; Lugão, A.B. Thermal, tensile and rheological properties of high density polyethylene (HDPE) processed and irradiated by gamma-ray in different atmospheres. AIP Conf. Proc. 2014, 1593, 236–239.
- Lee, M.J.; Ong, C.S.; Lau, W.J.; Ng, B.C.; Ismail, A.F.; Lai, S.O. Degradation of PVDF-based composite membrane and its impacts on membrane intrinsic and separation properties. Polym. Eng. 2016, 36, 261–268.
- Oshima, A.; Tabata, Y.; Kudoh, H.; Seguchi, T. Radiation induced crosslinking of polytetrafluoroethylene. Phys. Chem. 1995, 45, 269–273.
Since this study is a comparative study on the oxidation reaction of three polymers under the same conditions, the mechanical properties will be presented in a future study.
Point 3. Thermal properties are also missing. Authors must study the behavior of the polymers before and after irradiation.
Response 3:
We have agreed with your opinion. We already wrote about the thermal properties of the polymer in the paper at line 209. We put the sentences below. The reason for the drop in Td and Tm has been explained in the text, and the following references have been added to improve readers' understanding.
“Thermal properties related to the degradation could be attributed to structural de-composition (of main chain) due to oxidation [50-52]. Therefore, among the polymers analyzed, PTFE, with the highest fluorine content, underwent the least oxidation and decomposition by gamma-ray irradiation, and exhibited the highest radiation resistance.”
- Tamada, M. Radiation processing of polymers and its applications. Radiation applications. Springer, Singapore, 2018; 63–80.
- Medeiros, A.S.; Gual, M.R.; Pereira, C.; Faria, L.O. Thermal analysis for study of the gamma radiation effects in poly (vinylidene fluoride). Phys. Chem. 2015, 116, 345–348.
- Ewais, A.M.R.; Rowe, R.K. Effect of aging on the stress crack resistance of an HDPE geomembrane. Degrad. Stab. 2014, 109, 194–208.

Round 2
Reviewer 1 Report
The authors have well revised the manuscripts.
Author Response
Dear reviewer,
We appreciate for positive opinions of the reviewer.
Sincerely,
Reviewer 2 Report
In the previous revision I wrote:
"This manuscript reports on the study of the oxidation induced by gamma ray irradiation on different polymers. Irradiation with gamma ray of polymers (and fluoropolymers) is not new so it should be clearly stated which is the novelty of this work because it is not clear. At the same time, experimental results should be compared to literature.Besides this important point other aspects to be clarified/improved are..."
The manuscript has not been modified related to this point in the introduction/motivation part and results have not been discussed in comparison to literature. Additionally, for the rest of the questions, the authors answered to them but they did not include the information in the revised manuscript. It is not that I need the answer but the information needs to be clarified and completed since it is of interest for the readers and anyone could wonder about the same questions. So my recommendation is the same.
Author Response
Dear reviewer,
We greatly appreciate the reviewer’s time and efforts in improving our manuscript. We thank the reviewer for pointing out this error, which has been corrected in revision at line 63-70.
Sincerely,
Reviewer 3 Report
The author's replies are satisfactory.
However, DSC is not sufficient to study the thermal properties. TGA must be reported before possible publication.
Author Response
Dear reviewer,
We strongly agreed with your comment. As you said, reporting the TGA is important. Thus, we have already reported the derivative of TGA, DTG, in Figure 5(a)-(c).
We greatly appreciate the reviewer’s time and efforts in improving our manuscript.
Sincerely,
Round 3
Reviewer 2 Report
There are no additional comments